# Association between Advanced Glycation End-Products and Sarcopenia in Patients with Chronic Kidney Disease

**DOI:** 10.3390/biomedicines10071489

**Published:** 2022-06-23

**Authors:** Paolo Molinari, Lara Caldiroli, Elena Dozio, Roberta Rigolini, Paola Giubbilini, Massimiliano M. Corsi Romanelli, Giuseppe Castellano, Simone Vettoretti

**Affiliations:** 1Unit of Nephrology, Dialysis and Kidney Transplantation, Fondazione IRCCS Ca’ Granda Ospedale Maggiore Policlinico di Milano, 20122 Milan, Italy; paolo.molinari1@unimi.it (P.M.); lara.caldiroli@policlinico.mi.it (L.C.); giuseppe.castellano@unimi.it (G.C.); 2Department of Biomedical Science for Health, Laboratory of Clinical Pathology, Università degli Studi di Milano, 20133 Milan, Italy; elena.dozio@unimi.it (E.D.); mmcorsi@unimi.it (M.M.C.R.); 3Service of Laboratory Medicine1-Clinical Pathology, IRCCS Policlinico San Donato, San Donato Milanese, 20097 Milan, Italy; roberta.rigolini@grupposandonato.it (R.R.); paola.giubbilini@grupposandonato.it (P.G.); 4Department of Clinical Sciences and Community Health, Università degli Studi di Milano, 20122 Milan, Italy

**Keywords:** advanced glycation end-products (AGEs), chronic kidney disease (CKD), sarcopenia, soluble receptor for AGE (sRAGE), cleaved RAGE (cRAGE), endogenous secretory RAGE (esRAGE)

## Abstract

Background: In patients with chronic kidney disease (CKD), there is an overproduction and accumulation of advanced glycation end-products (AGEs). Since AGEs may have detrimental effects on muscular trophism and performance, we evaluated whether they may contribute to the onset of sarcopenia in CKD patients. Methods: We enrolled 117 patients. The AGEs were quantified by fluorescence intensity using a fluorescence spectrophotometer and soluble receptor for AGE (sRAGE) isoforms by ELISA. As for the sarcopenia definition, we used the European Working Group on Sarcopenia in Older People (EWGSOP2) criteria. Results: The average age was 80 ± 11 years, 70% were males, and the mean eGFR was 25 + 11 mL/min/1.73 m^2^. Sarcopenia was diagnosed in 26 patients (with a prevalence of 22%). The sarcopenic patients had higher levels of circulating AGEs (3405 ± 951 vs. 2912 ± 722 A.U., *p* = 0.005). AGEs were higher in subjects with a lower midarm muscle circumference (MAMC) (3322 ± 919 vs. 2883 ± 700 A.U., respectively; *p* = 0.005) and were directly correlated with the gait test time (r = 0.180, *p* = 0.049). The total sRAGE and its different isoforms (esRAGE and cRAGE) did not differ in patients with or without sarcopenia. Conclusions: In older CKD patients, AGEs, but not sRAGE, are associated with the presence of sarcopenia. Therefore, AGEs may contribute to the complex pathophysiology leading to the development of sarcopenia in CKD patients.

## 1. Introduction

Sarcopenia is a pathological entity characterized by a progressive reduction of skeletal muscle mass, strength, and function [1]. Even though sarcopenia is mainly correlated with aging, it can be developed in different pathological conditions independently of age. Among different disorders, chronic kidney disease (CKD) can promote premature aging [2] and play a prominent role in sarcopenia onset [3]. In fact, sarcopenia is more prevalent at lower levels of renal function [4].

In CKD, different conditions, such as chronic inflammation, insulin resistance, and the accumulation of uremic toxins, can contribute to the risk of developing age-related disorders, including sarcopenia [4,5,6].

Advanced glycation end-products (AGEs) derive from non-enzymatic modifications of amino groups of proteins or lipids by reducing sugars and their metabolites through the Maillard reaction and polyol pathway [1]. Several environmental factors, such as cigarette smoke, a diet rich in carbohydrates, hypercaloric diets, highly-processed foods, and a sedentary lifestyle, can induce AGE production [7]. AGEs, through the promotion of oxidative stress, lead to the activation of several stress-induced transcription factors with the production of pro-inflammatory mediators [7]. More than 20 different AGEs have been identified, both in food and in human serum, and they can be divided into fluorescent and non-fluorescent. Although they are characterized by diverse chemical structures, they share the presence of a lysine residue in their molecules [8,9]. In CKD, the accumulation of AGEs is due both to the reduced renal clearance and increased production, which is the net result of an imbalance between oxidant/antioxidant homeostasis. Therefore, AGEs may be considered uremic toxins [10,11,12].

In patients with CKD, the binding of AGEs with their membrane receptor (RAGE) has been demonstrated to cause oxidative stress and inflammation and promote insulin resistance and endothelial dysfunction [12,13,14]. RAGE is a transmembrane protein belonging to the immunoglobulin superfamily. Higher levels of this receptor are found in pathological conditions characterized by chronic inflammation [15]. The binding of ligands to RAGE induces the activation of several signaling pathways, which, in turn, lead to the activation of the transcription factor, NF-Kβ, which causes an increased production of pro-inflammatory mediators [15]. Furthermore, RAGE’s activation causes the activation of MAPK, JAK-STAT, and phosphoinositol 3 kinase, subsequently leading to inflammatory, proliferative, angiogenic, fibrotic, thrombogenic, and apoptotic reactions [16]. Moreover, the binding of AGEs to their receptor increases the production of a reactive oxygen species (ROS) through the activation of an NADPH oxidase and several mitochondrial pathways [17]. Therefore, AGEs’ accumulation contributes to the development of several complications associated with CKD, such as an increased burden of atherosclerosis, cardiovascular disease, and anemia, and can further promote CKD’s progression [18,19]. sRAGE is the soluble circulating form of RAGE. sRAGE is a pool composed of the endogenously secretory form (esRAGE) that is considered the real decoy receptor and the membrane-cleaved form (cRAGE), which is considered a surrogate marker of inflammation [1]. The role of these soluble isoforms of RAGE is still to be defined. In fact, on the one hand, some studies have pointed towards a possible protective effect of sRAGE, exerted by binding circulating AGEs, thus favoring clearance of these molecules [20]. On the other hand other studies showed that sRAGE acts more as a marker of increased RAGE production and activity, rather than having a protective effect [21].

In the elderly, increased concentrations of AGEs are associated with poor handgrip strength and a low walking speed, two dysfunctions that are strongly correlated with sarcopenia [22,23]. Moreover, in vitro studies indicated that AGEs could cause muscle atrophy and impair myogenesis through a RAGE-mediated signaling pathway [24].

In this context, we hypothesize that the accumulation of AGEs that characterizes the advanced stages of CKD might contribute to the pathogenesis of sarcopenia in this population. Therefore, in this observational study, we explored whether in patients affected by the advanced CKD serum AGEs and the different RAGE isoforms of soluble RAGE, i.e., the cleaved RAGE (cRAGE), and the endogenous secretory RAGE (esRAGE), are associated with sarcopenia.

## 2. Materials and Methods

### 2.1. Patients and Study Design

We performed a cross-sectional study, enrolling 117 prevalent patients attending our outpatient CKD clinic between 9/2016 and 3/2018. Selection criteria were: age ≥ 65 years, CKD stages 3a to 5, not yet on dialysis in conservative therapy, and a relatively stable eGFR over the previous 6 months (with less than 2 mL/min/1.73/m^2^ of variation). eGFR was estimated according to the CKD-EPI formula. To exclude possible confounding factors, we excluded patients with pro-inflammatory conditions, such as cancer, cirrhosis and/or ascites, severe heart failure (NYHA class III–IV), nephrotic syndrome, thyroid diseases, bowel inflammatory diseases, and inability to cooperate. We also excluded patients treated with immunosuppressive drugs or who had been hospitalized in the last three months. Urinary and biochemical parameters were collected on the same visit, in the morning, and after an overnight fast of at least 12 h. The study was conducted according to the ICP Good Clinical Practices Guidelines, and it was approved by the Ethics Committee of our institution (Milano 2-approval N. 347/2010). All patients signed an informed consent.

### 2.2. sRAGE, esRAGE, and cRAGE Quantification

The total sRAGE and its isoforms were quantified as previously indicated [25]. In brief, sRAGE and esRAGE were measured using two ELISA kits, respectively, from R&D Systems (DY1145, Minneapolis, MN, USA) and from B-Bridged International (K1009–1, Santa Clara, CA, USA). The intra- and inter-assay coefficients of the variation of the esRAGE assay were 6.37% and 4.78–8.97%, respectively. cRAGE levels were obtained by subtracting the esRAGE from the total sRAGE. The AGE/sRAGE ratio was then obtained. Photometric measurements were obtained using the GloMax®-Multi Microplate Multimode Reader (Promega, Milan, Italy).

### 2.3. AGE Quantification

We quantified AGEs by measuring using a fluorescence spectrophotometer (The GloMax^®^, Promega, Milano, Italy); the fluorescence intensity of the plasma samples was at 414–445 nm after excitation at 365 nm, as previously reported [26,27]. Fluorescence was expressed as the relative fluorescence intensity in arbitrary units (AU). AGEs were then normalized for total protein content.

### 2.4. Assessment of Sarcopenia

As already described [28], for the sarcopenia definition, we used the European Working Group on Sarcopenia in Older People (EWGSOP2) criteria as the presence of low muscle strength and reduced muscle quantity [29].

Muscle strength was assessed using a Jamar dynamometer (Sammons Preston Inc., Bolingbrook, IL, USA) to measure the handgrip strength values; <16 kg in females and <27 kg in males were considered reduced [29].

Muscle mass was considered reduced when the reduction of mid-arm muscle circumference (MAMC) was >10% in relation to the fiftieth percentile of the reference population [29].

Patients were considered severe sarcopenic if they also had a reduced speed (<0.8 m/s) at the 4 m gait speed test.

### 2.5. Anthropometric Measurements

We measured body weight, height, body mass index (BMI, calculated according to Quetelet Index (kg/m^2^)), and mid-arm circumference (MAC) while, with a Harpenden skinfold caliper, we measured the tricipital and bicipital skinfold (TST; BST) on the dominant arm. MAMC was derived from MAC and TST as follows: MAMC (cm) = AC (cm) − (πxTSF (cm)).

### 2.6. Statistical Analysis

We expressed continuous variables as the mean with the standard deviation (SD) in parametric distributions, or the median with an interquartile range (IQR) in non-parametric data. Categorical variables were summarized as percentages. We performed the Student’s *t*-test and ANOVA to compare parametric variables, while we performed, when appropriate, the Mann–Whitney “U” test, or the Kruskal–Wallis, for the comparison of the non-parametric ones.

The general linear model (GLM) was used to test for the correlation between sarcopenia domains, AGEs, and RAGEs’ isoforms.

Statistical analysis was conducted using IBM SPSS software (version 25).

## 3. Results

### 3.1. Population Characteristics

The anthropometry and clinical and metabolic characteristics are depicted in Table 1. The mean age of our cohort was of 80 years, with a marked male preponderance. Nearly half of the examined patients suffered from diabetes, and almost all were hypertensive. The mean eGFR was around 25 mL/min/1.73 m^2^, while the measured creatinine clearance was 28 mL/min. Our cohort was averagely overweight, with a mean BMI of 28 kg/m^2^ (Table 1). The metabolic markers settled down on average around the normal values, except for uric acid and CRP, which were near the upper limit of normality.

Sarcopenia was diagnosed in 26 patients, approximately 22% of the overall cohort. Regarding the comparison between sarcopenic (Src) and non-sarcopenic (N-Src) patients, we observed that the Src patients were significantly older (*p* = 0.001) and had a lower BMI (*p* < 0.0001). The Src patients had a lower creatinine clearance (*p* = 0.039), although the difference in the eGFR between the two groups was not significant.

Among the metabolic parameters that were examined, the only significant difference between the Src and N-Src patients was relative to the CRP, which was higher in Src patients (*p* = 0.003).

### 3.2. Advanced Glycation End-Products

We compared the concentrations of AGEs and the different sRAGE isoforms in the Src and N-Src individuals. As depicted in Table 2, we found out that the serum AGE’s concentration was significantly higher in the Src patients, even after correcting for the eGFR (*p* = 0.02). Moreover, we found higher AGE concentration in subjects with reduced MAMC in our cohort and also, in this case, the statistical significance persisted even after correction for the eGFR (*p* = 0.049, Table 3).

Overall, we did not find any correlation between the concentrations of the different soluble RAGE isoforms and the presence of sarcopenia, even after testing them separately in the different domains that contribute to defining the presence and severity of sarcopenia. (Table 2 and Table 3).

In order to investigate the interplay between AGEs and the other variables associated with sarcopenia, we performed a multivariate analysis (Table 4) where we included the factors that showed the strongest correlations with sarcopenia. In this model, the AGE concentration was not independently associated with the presence of sarcopenia.

Finally, as shown in Figure 1, we performed two linear regression models, exploring the potential association of the AGE concentrations with individual handgrip strength and gait speed performances. Concerning handgrip performance, we found out that AGEs concentrations were inversely associated with patients’ handgrip strength, even if it did not reach statistical significance. Furthermore, we found a direct and significant association between the AGE concentrations and patients’ gait test time.

We also performed an analysis to evaluate whether there was any potential difference in the AGE and sRAGE concentrations between diabetic and non-diabetic patients. However, there were no significant differences between the two groups (Appendix A).

## 4. Discussion

The main result of our study was the observation that in older patients affected by advanced CKD, those affected by sarcopenia have the highest levels of circulating AGEs. The association between AGEs and sarcopenia was further confirmed by the fact that at decreasing MAMC and muscular function, evaluated as the gait time test, AGEs increase.

AGEs are uremic toxins that accumulate alongside the progression of renal impairment [30,31]. In CKD patients, serum AGE concentrations are associated with complications, such as cardiovascular disease, heart failure, and anemia, [18,19,32] as well as with higher all-cause mortality [33]. In older individuals not selected for CKD, increased AGEs have been associated with features of sarcopenia, such as reduced grip strength and a slower gait speed [23,24]. AGEs are uremic toxins that accumulate alongside the progression of renal impairment [30]. In CKD patients, serum AGE concentrations are associated with complications, such as cardiovascular disease, heart failure, and anemia, [18,19,24] as well as with higher all-cause mortality [31]. In older individuals not selected for CKD, increased AGEs have been associated with features of sarcopenia, such as reduced grip strength and a slower gait speed [22,23]. Our study confirmed these results also in patients affected by CKD whose AGE concentrations are known to be higher than in the general population [10]. In particular, in our study, higher AGE levels were significantly correlated with a lower MAMC and a lower gait speed, which may be considered as a proxy of reduced overall muscular functionality.

Sarcopenia is the age-related loss of skeletal muscle mass and strength. It occurs with aging, but it can be observed in many different pathological conditions, such as CKD, and it has attracted attention for its association with a functional disability over the years [32,33]. The onset of sarcopenia is a complex phenomenon where the balance between protein synthesis and degradation is responsible for the maintenance of muscle mass. Therefore, muscle mass can be affected by any condition, increasing protein degradation or decreasing protein synthesis [10]. AGEs can play a role in the onset and progression of sarcopenia also in CKD patients [1]. Considering the different domains of sarcopenia, the detrimental influence of AGEs on MAMC is supported by some basic research studies, suggesting that AGEs may induce muscle atrophy by acting directly on myogenesis and muscle regeneration, thus [24]. Therefore, it is likely that one of the first noticeable effects of high AGE concentrations at the muscular level may be the reduction of the overall muscle mass.

More recently, it was demonstrated that the accumulation of AGEs in mice’s muscular tissue might induce morphological abnormalities, such as capillary rarefaction and mitochondrial dysfunction [6]. Thus, AGEs may contribute to the onset of sarcopenia by reducing muscular trophism, as well as by impairing muscular performance (since they may affect both mitochondrial activity and cellular metabolism) [34]. In particular, AGEs contribute to impaired mitochondrial energy production by reducing succinate dehydrogenase (SDH) activity and peroxisome proliferator-activated receptor gamma coactivator 1-alpha (PGC1-α) levels [35,36]. AGEs can also negatively impact muscle function by reducing the turnover of the extracellular matrix and by increasing the stiffness of muscular connective tissue [37]. Furthermore, the detrimental effects of AGEs on muscular tissue seem to be also mediated by their interaction with RAGE, which is generally upregulated in muscles with high levels of AGE deposition [6,38]. RAGE activation may promote the onset of muscular dysfunction by lowering the expression of myogenin [24,39]. Moreover, the activation of the AGE–RAGE pathway seems to impair also muscular fiber distribution, reducing the overall strength developed by cellular contraction. Finally, in a recent study, the higher deposition of AGEs in patients’ skin (evaluated by autofluorescence) correlates with sarcopenia [40].

In our study, in the multivariate analysis, sarcopenia maintained an independent correlation only with age, BMI, and CRP. Various studies found that higher AGE concentrations were associated with weight loss and reduced fat mass [6]. This may depend on the fact that AGEs’ accumulation provokes an increase in inflammation, which enhances protein catabolism and increases resting energy expenditure, thus inducing a global catabolic status [41]. The upregulation of pro-inflammatory markers is well known to act as the pathophysiological basis underpinning several disorders, including sarcopenia [42]. Several molecular pathways may link systemic inflammation to muscle wasting by inducing an imbalance between protein synthesis and catabolism [43,44], and high levels of inflammatory cytokines have been related to lower muscle strength and mass [45,46]. An association between AGEs and inflammation has also been found in subjects undergoing chronic hemodialysis [47]. AGEs can activate the pro-inflammatory cascade via RAGEs both systemically and locally in muscular tissue [48]. All these assumptions could explain why, in the multivariate analysis, the association between AGEs and sarcopenia lost its strength when compared to BMI and CRP. Since AGEs are involved in both the pathophysiological mechanisms that increase inflammatory response and malnutrition, it is possible that their independent contribution to the development of muscle wasting sarcopenia is less evident in the advanced stages of CKD, in which inflammation and malnutrition have been already persisting since a long time. Therefore, we believe that the loss of significance in the multivariate analysis does not reduce the relevance of the association between AGEs and sarcopenia in patients with advanced CKD. Conversely, we believe that our results confirm that AGEs’ accumulation may contribute to impaired muscle health and functionality and suggest that this effect may also be mediated also by a more complex interplay with nutritional status and systemic inflammation.

The total sRAGE and its different isoforms, esRAGE and cRAGE, have also been evaluated in this study as potential markers of sarcopenia. Conflicting results exist about the association of the total sRAGE with sarcopenia. In one study, the total sRAGE was shown to determine muscle mass and fat mass in hemodialysis patients [49], whereas, in another study by Fukasawa et al., sRAGE was of limited significance in identifying malnutrition [49]. Although these previous results have been obtained in a different population, namely the hemodialysis patients, the results we obtained in advanced CKD patients seem to be in line with data described by Fukasawa et al., To our knowledge, our study was the first to explore the role of sRAGE and its isoforms as biomarkers of sarcopenia in advanced CKD. Additional studies are therefore needed to confirm our observation.

We acknowledge that our study has some limitations. First of all, the retrospective cross-sectional nature of our work does not allow us to better evaluate the potential independent association of AGE levels on the development of sarcopenia. Therefore, our results can be considered only preliminary and “hypothesis generating” about the complex interplay between AGEs’ accumulation and the development of sarcopenia in older individuals affected by advanced CKD. Secondly, our study is monocentric, and our population is relatively small. However, the monocentric nature of our study allowed us to reduce the possible sources of bias by using a highly standardized protocol for biochemical analyses and clinical observations. In particular, we applied strict inclusion and exclusion criteria that allowed us to exclude patients that may have developed muscular wasting and sarcopenia because of specific clinical conditions.

A significant point of strength of our study is that the correlation between AGEs and sarcopenia was thoroughly investigated. In fact, our analyses focused on many factors that could be related to or influence the relationship with sarcopenia in some way. Furthermore, we studied the in-depth association of AGEs and sarcopenia by addressing not only AGEs but also sRAGE isoforms. This is, to the best of our knowledge, the first time that a comprehensive evaluation of the association between the AGEs–RAGE system and sarcopenia has been conducted in humans.

In conclusion, we found that AGEs but not sRAGE and its isoforms are associated with sarcopenia in older patients with advanced CKD. This association may be at least in part mediated by the influence of AGEs on systemic inflammation and malnutrition. Whether these preliminary observational results were also confirmed by prospective studies, AGEs may become a target for specific interventions aimed at reducing the onset of sarcopenia in CKD patients.

## Figures and Tables

**Figure 1 biomedicines-10-01489-f001:**
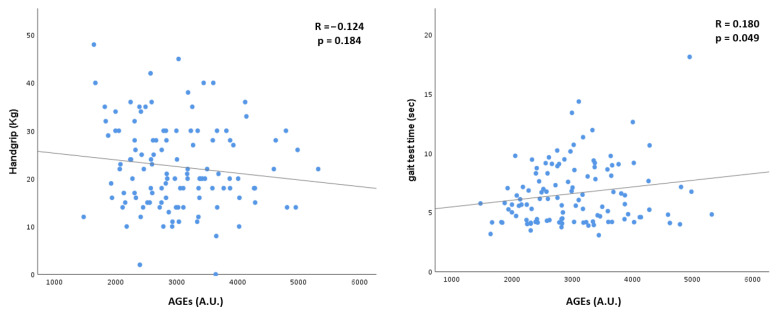
Linear regression model of comparison between AGEs, handgrip strength, and gait test time; AGEs, Advanced Glycation End-products.

**Table 1 biomedicines-10-01489-t001:** Patients’ cohort characteristics.

Variables	Overall Cohort (*n* = 117)	N-Src (*n* = 91)	Src (*n* = 26)	*p*
*General characteristics*				
Age, (years)	80 ± 11	76 ± 12	83 ± 6	**0.001**
Males, n (%)	82 (70)	62 (68)	20 (77)	0.28
Diabetes, n (%)	65 (56)	52 (65)	13 (50)	0.65
Hypertension, n (%)	104 (89)	81 (89)	23 (88.5)	0.91
BMI, (kg/m^2^)	28 ± 5	28.6 ± 4.8	24.8 ± 3.9	**<0.0001**
eGFR, (mL/min/1.73 m^2^)	25 ± 11	26 ± 11	22 ± 8	0.08
Creatinine clearance, (mL/min/1.73 m^2^)	28 ± 16	30 ± 17	23 ± 13	**0.039**
*Metabolic characteristics*				
Uric Acid, (mg/dL)	6 ± 1.5	6.1 ± 1.4	6 ± 1.8	0.42
HbA1c, (mmol/dL)	47 ± 11	47 ± 10	48 ± 15	0.83
Total Cholesterol, (mg/dL)	168 ± 37	167 ± 36	173 ± 39	0.84
Albumin, (g/dL)	4 ± 0.4	4.0 ± 0.3	4.1 ± 0.5	0.43
Prealbumin, (mg/dL)	28 ± 5	29 ± 5	27 ± 6	0.12
CRP, (mg/dL)	0.4 ± 0.7	0.3 ± 0.4	0.8 ± 1.3	**0.003**

eGFR, estimated glomerular filtration rate; BMI, Body mass index; HbA1c: glycated hemoglobin; CRP, c-reactive protein; N-Src, non-sarcopenic patients; Src, sarcopenic patients. Data are expressed as the mean with the standard deviation or as a number and percentages. *p* values less than 0.05 are indicated in bold.

**Table 2 biomedicines-10-01489-t002:** Concentration of AGEs and sRAGE isoforms in sarcopenic and non-sarcopenic CKD patients.

Variables	N-Src (*n* = 91)	Src (*n* = 26)	*p*	*p* (eGFR Weighted)
AGEs (arbitrary unit)	2912 ± 722	3405 ± 951	**0.005**	**0.02**
sRAGE (pg/mL)	2338 ± 1280	2411 ± 1268	0.86	0.63
esRAGE (pg/mL)	656 ± 503	713 ± 448	0.60	0.96
cRAGE (pg/mL)	1693 ± 937	1698 ± 902	0.93	0.51
AGEs/sRAGE (arbitrary unit)	1.59 ± 0.89	1.8 ± 1.2	0.23	0.19

AGEs, Advanced Glycation End-products; sRAGE, soluble receptor for AGE; esRAGE: endogenous secretory receptor for AGE; cRAGE: cleaved receptor for AGE; CKD, chronic kidney disease; N-Src, non-sarcopenic patients; Src, sarcopenic patients. Data are expressed as the mean with the standard deviation. *p* values less than 0.05 are indicated in bold.

**Table 3 biomedicines-10-01489-t003:** Concentration of AGEs and sRAGE isoforms classified according to the presence (yes) or absence (not) of alterations in the sarcopenic domains.

Variables	Yes	Not	*p*	*p*
				(eGFR Weighted)
*Reduced MAMC, n (%)*	*37 (31)*	*80 (69)*	** *<0.0001* **	** *<0.0001* **
AGEs (arbitrary unit)	3322 ± 919	2883 ± 700	0.005	0.049
sRAGE (pg/mL)	2426 ± 1292	2350 ± 1287	0.77	0.58
esRAGE (pg/mL)	536 (398–707)	552 (368–787)	0.66	0.75
cRAGE (pg/mL)	1727 ± 921	1707 ± 959	0.91	0.53
AGEs/sRAGE (arbitrary unit)	1.8 ± 1.1	1.6 ± 0.9	0.25	0.19
*Reduced Gait Speed Test, n (%)*	*76 (64)*	*41 (36)*	** *<0.0001* **	** *0.001* **
AGEs (arbitrary unit)	2977 ± 750	3101 ± 882	0.42	0.26
sRAGE (pg/mL)	2373 ± 1310	2376 ± 1249	0.99	0.90
esRAGE (pg/mL)	630 (368–776)	526 (390–720)	0.75	0.66
cRAGE (pg/mL)	1728 ± 972	1687 ± 902	0.82	0.89
AGEs/sRAGE (arbitrary unit)	1.6 ± 0.9	1.6 ± 1	0.94	0.93
*Reduced handgrip strength, n (%)*	*68 (58)*	*49 (42)*	** *<0.0001* **	** *<0.0001* **
AGEs (arbitrary unit)	3054 ± 809	2976 ± 787	0.60	0.78
sRAGE (pg/mL)	2343 ± 1330	2416 ± 1230	0.76	0.39
esRAGE (pg/mL)	543 (367–718)	515 (384–787)	0.50	0.84
cRAGE (pg/mL)	1662 ± 937	1782 ± 958	0.50	0.23
AGEs/sRAGE (arbitrary unit)	1.7 ± 1	1.5 ± 0.9	0.34	0.30

AGEs, Advanced Glycation End-products; sRAGE, soluble receptor for AGE; esRAGE: endogenous secretory receptor for AGE; cRAGE: cleaved receptor for AGE; CKD, chronic kidney disease; MAMC, mid-arm muscle circumference. *p* values less than 0.05 are indicated in bold.

**Table 4 biomedicines-10-01489-t004:** Multivariate analysis of the relationship between AGEs, eGFR, age, CRP, BMI, and sarcopenia.

Variables	Odds Ratio	Odds Ratio (CI)	*p*
AGEs (arbitrary unit)	1.5	(0.91–2.19)	0.19
eGFR, (mL/min/1.73 m^2^)	0.99	(0.93–1.06)	0.84
Age, (years)	1.11	(1.03–1.2)	**0.008**
CRP (mg/dL)	2.2	(1.08–4.5)	**0.029**
BMI (kg/m^2^)	0.77	(0.66–0.9)	**0.001**

AGEs, Advanced Glycation End-products; estimated glomerular filtration rate (eGFR); CRP, C-reactive protein; BMI, body mass index. *p* value; *p* values less than 0.05 are indicated in bold.

## Data Availability

The dataset analyzed for this study can be found in the OSF repository at https://osf.io/7zxh6/?view_only=47bd43af3a0946b08280ef90b388782d. Accessed on 27 March 2022.

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
