# Peer review of "Association between Advanced Glycation End-Products and Sarcopenia in Patients with Chronic Kidney Disease"

_biomedicines, 2022, doi:10.3390/biomedicines10071489_

Round 1

Reviewer 1 Report

A well done study to identify correlation of AGE and sarcopenia. Would have liked to have some reference about why sRAGE levels are low in long lived individuals but in this study with CKD patients are significantly elevated in older aged individuals 

Reviewer 2 Report

The manuscript "Association between advanced glycation end-products and sarcopenia in patients
with chronic kidney disease" contains very important information about a human disease common, which it was taken
scientifically and bioethical sense, and described understandably their results. Also, the discussion section put emphasis
that with the present results it was generated another hypothesis starting the participation of AGEs with sarcopenia in individuals.

Reviewer 3 Report

Although the study is valuable, it has some shortcomings. Various situations should be considered that will increase the research value. The introduction and method section should be modified with clear understanding for readers and should be rearranged to be more understandable.

Typos should be corrected. The article should be accepted after minor revision.
